# PolyNODE: Variable-dimension Neural ODEs on M-polyfolds

## Abstract

Neural ordinary differential equations (NODEs) are geometric deep learning models based on dynamical systems and flows generated by vector fields on manifolds. Despite numerous successful applications, particularly within the flow matching paradigm, all existing NODE models are fundamentally constrained to fixed-dimensional dynamics by the intrinsic nature of the manifold's dimension. In this paper, we extend NODEs to M-polyfolds (spaces that can simultaneously accommodate varying dimensions and a notion of differentiability) and introduce PolyNODEs, the first intrinsic variable-dimensional flow-based model in geometric deep learning. As an example application, we construct explicit M-polyfolds featuring dimensional bottlenecks and PolyNODE autoencoders based on parametrised vector fields that traverse these bottlenecks. We demonstrate experimentally that our PolyNODE models can be trained to solve reconstruction tasks in these spaces, and that latent representations of the input can be extracted. The code used in our experiments is publicly available at [omitted for reviewing].

## 1 Introduction

Many modern neural networks change the dimension of their hidden representation across layers. They expand features to increase expressive power or compress them to extract information-dense latent representations. The ability to accommodate variable feature space dimensions is central in encoder-decoder models, convolutional neural networks, and many other ubiquitous feed-forward machine learning architectures. A natural question is: what happens if we take the continuous-depth viewpoint in this setting? What is a continuous-depth model whose state dimension is allowed to change over time?

Neural ordinary differential equations (NODEs) provide the continuous-depth limit of residual-type networks with constant-width layers by replacing discrete layers with the flow of an ordinary differential equation (ODE) (Chen et al., 2018). In its basic form, the hidden state evolves as

$$\frac{d}{dt}\phi(t,p) = X(\phi(t,p)), \qquad \phi(0,p) = p, \tag{1}$$

where $X : \mathbb{R}^n \to \mathbb{R}^n$ is the vector field generating the flow $\phi : \mathbb{R} \times \mathbb{R}^n \to \mathbb{R}^n$ and $h(p) = \phi(1,p)$ is the diffeomorphism defining the map between the input and the output of the NODE model. This construction extends to manifold-valued states by replacing $\mathbb{R}^n$ with a manifold $M$ and generalising to flows $\phi : \mathbb{R} \times M \to M$ generated by vector fields $X$ on $M$ (Falorsi & Forré, 2020; Lou et al., 2020; Mathieu & Nickel, 2020). NODEs are widely used in flow-based learning and generative modelling, and features attractive properties such as universality (Zhang et al., 2020; Andersdotter et al., 2025) and scalable training using flow matching (Liu et al., 2023; Lipman et al., 2023; Albergo & Vanden-Eijnden, 2023; Tong et al., 2024).

In (1), the state $\phi(t,p)$ always lies in a space of fixed dimension, either $\mathbb{R}^n$ or a manifold $M$. This is not a limitation of parametrisation but an intrinsic geometric restriction. A smooth manifold does not allow trajectories whose number of local degrees of freedom drops or increases at an intermediate time. For this reason, NODEs do not provide an intrinsic continuous-depth analogue of, e.g., encoder-decoder architectures (LeCun, 1987; Bourlard & Kamp, 1988; Baldi & Hornik, 1989; Hinton, 1989) where a dimensional bottleneck is used to extract a latent representation from which the original data can be efficiently reconstructed.

More generally, NODEs cannot capture variable-width feed-forward network dynamics in continuous time (Ruiz-Balet & Zuazua, 2023).

If we insist on changing feature dimensions, then the correct state space is no longer a manifold. The continuous time analogue is instead a stratified space, where smooth pieces of different dimensions meet along singular loci. A simple example of such a space corresponding to the dimensional bottleneck of the autoencoder is $\Omega_n^m := ((-\infty, \tau_1) \times \mathbb{R}^n \times \mathbb{R}^m) \cup ([\tau_1, \tau_2] \times \mathbb{R}^n \times \{0\}) \cup ((\tau_2, \infty) \times \mathbb{R}^n \times \mathbb{R}^m)$. Outside $[\tau_1, \tau_2]$ trajectories evolve in $(\mathbb{R} \setminus [\tau_1, \tau_2]) \times \mathbb{R}^n \times \mathbb{R}^m$, but on $[\tau_1, \tau_2]$ they are constrained to $[\tau_1, \tau_2] \times \mathbb{R}^n \times \{0\}$. At $\tau_1$ and $\tau_2$, the local dimension changes, and at these points there are no manifold charts since the space is not locally homeomorphic to $\mathbb{R}^n$. Consequently, the standard manifold calculus that underlies NODE theory breaks down.

A natural approach to extend NODEs would be to embed $\Omega_n^m$ into an ambient Euclidean space and define an ordinary NODE there. The ambient space approach is explored in augmented NODEs, described for the Euclidean case in (Dupont et al., 2019) and for general manifolds in (Andersdotter et al., 2025). A related construction is the AutoencODEs presented in (Cipriani et al., 2025), which uses explicitly discontinuous vector fields to achieve dimensional change by deactivating the dynamics in subsets of the ambient coordinates. However, neither approach resolves the main issue, which is the calculus at the singular set. If we require the ambient vector field to be Lipschitz and tangent to the stratified space, then its flow is necessarily injective and therefore unable to probe a genuine bottleneck in finite time. This would, for example, be the case for a vector field realised by a generic sequential neural network with piecewise smooth activation functions. Finite time collapse of coordinates forces the dynamics to become singular or at least non-Lipschitz near the transition, and then the usual sensitivity calculus used for training NODEs is no longer justified. What we need is an intrinsic notion of smoothness on the stratified state space that keeps a chain rule and a well-defined tangent map while still allowing controlled collapse.

M-polyfold theory provides exactly this (Hofer et al., 2007; 2009a;b; Weber, 2019; Hofer et al., 2021; Åhag et al., 2026). It was developed to give a workable differential calculus on spaces that arise from glueing and degeneration, where effective dimension changes are part of the geometry. In our setting, the key point is that an M-polyfold equips a stratified space with a global notion of differentiability that records how the collapsing coordinates vanish as one approaches the bottleneck. This replaces the missing manifold structure at $\tau_1$ and $\tau_2$ and restores the tools needed for flow-based learning, including a chain rule and a tangent construction that are compatible with the bottleneck.

In this paper, we use the M-polyfold framework to extend NODEs beyond manifolds. We introduce PolyN-ODEs, intrinsically variable-dimension continuous-depth models parametrised by vector fields on M-polyfolds. We explicitly construct M-polyfolds that realise dimensional bottlenecks and PolyNODE autoencoders that encode and decode by traversing these bottlenecks. We then show how standard backpropagation can be modified to accommodate the M-polyfold vector fields in practice, and demonstrate empirically that PolyNODE models can be trained to solve reconstruction tasks.

Our main contributions are as follows.

- We introduce PolyNODEs, a class of continuous-depth models on M-polyfolds that support intrinsic variable-dimension dynamics.

- We construct explicit M-polyfold bottleneck spaces and PolyNODE autoencoders by parametrising vector fields whose flows traverse the bottlenecks.

- We provide proof-of-concept experiments showing that PolyNODE autoencoders can be trained for reconstruction using backpropagation.

## 2 Neural ODEs on M-polyfolds

The description of a stratified space required to accommodate NODE state vectors that can change their dimension – its regular components, their boundaries and intersections – is quite cumbersome even for simple configurations like the space $\Omega_n^m$ introduced above and illustrated in Figure 1a. Generalising NODEs

to stratified spaces, the regular components correspond to regions of fixed dimension dynamics, and the boundaries to discontinuous changes in the dimension of the state vector $\phi(t, p)$, as illustrated in Figure 1b).

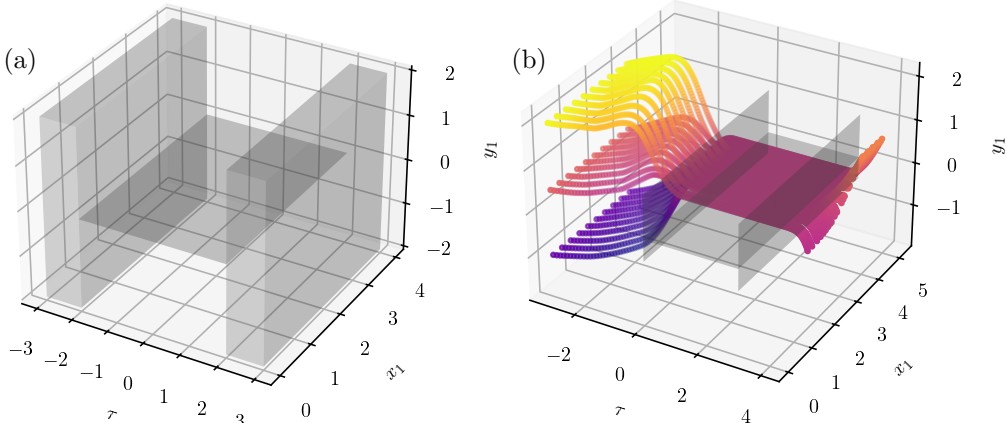

Figure 1: (a) Illustration (grey shading) of the stratified topological space $\Omega_1^1 \subset \mathbb{R}^3$ with $\tau_1 = -2$ and $\tau_2 = 2$. (b) Illustration of a semi-flow in $\Omega_1^1 \subset \mathbb{R}^3$ with $\tau_1 = -2$ and $\tau_2 = 2$, polyfold structure indicated as grey planes.

The concept of an M-polyfold allows for a consistent description of stratified spaces like $\Omega_n^m$. More importantly, polyfold theory endows these spaces with a global smooth structure, which gives us access to much the same analytical tools that are available on manifolds. In particular, we have well-defined notions of vector fields and can construct the flows required to extend NODEs to these spaces.

Below we present a very brief introduction to the theory of M-polyfolds based on (Hofer et al., 2021). Subsequently, we give an explicit M-polyfold structure for the stratified space $\Omega_n^m$, construct flows and ODEs on M-polyfolds, and define the PolyNODE model, which is our first main contribution. We use $\Omega_n^m$ as a recurring example, as it features in our construction of PolyNODE autoencoders in Section 3, and denote a point in this space as $p = (\tau, x, y)$ with $\tau \in \mathbb{R}$, $x \in \mathbb{R}^n$, and $y \in \mathbb{R}^m$.

## 2.1 Differential Geometry of M-polyfolds

**Scale calculus and retracts**  A *scale Banach space*, or sc-Banach space, is a Banach space $E$, together with a filtration $\{E_k\}_{k \in \mathbb{N}}$ called a *scale structure*, or sc-structure, where $E_0 = E$ and $E_{k+1} \subset E_k$ for all $k \in \mathbb{N}$, all inclusions $\iota_k : E_{k+1} \hookrightarrow E_k$ are compact, and the space $E_\infty := \bigcap_{k \in \mathbb{N}} E_k$ is dense in every *scale $E_k$*.

Each $E_k$ can be turned into a sc-Banach space of its own, denoted by $E^k$, where $(E^k)_i := E_{k+i}$ for all $i \in \mathbb{N}$. Direct sums of sc-Banach spaces are again sc-Banach spaces with component-wise scales. Note that finite-dimensional Banach spaces admit only the constant scale structure, i.e. $E_k = E$ for all $k \in \mathbb{N}$, since the only dense vector subspace of $E$ is $E$ itself. Infinite-dimensional Banach spaces, on the other hand, cannot be equipped with the constant scale structure.

In this article, we are mainly interested in the case of $L^2(\mathbb{R})$ with a weighted Sobolev sc-structure. Let $\{\delta_k\}_{k=0}^\infty$ be a non-negative, strictly increasing sequence with $\delta_0 = 0$, let $\alpha : \mathbb{R} \to [-1, 1]$ be a smooth odd function with $\alpha(0) = 0$ and $\alpha(s) = 1$ for $s > 1$. The weighted Sobolev norms are then given by

$$\|f\|_k^2 := \int_{\mathbb{R}} \sum_{i=0}^k |d^i f(s)|^2 e^{2\delta_k s \alpha(s)} \, ds \,,$$

and the scales of $L^2(\mathbb{R})$ are defined by

$$L^2(\mathbb{R})_k := \left\{ f \in L^2(\mathbb{R}) \,\middle|\, \|f\|_k < \infty \right\}. \tag{2}$$

The inclusion $\iota : L^2(\mathbb{R})_{k+1} \hookrightarrow L^2(\mathbb{R})_k$ is compact by embedding theorems for weighted Sobolev spaces, compare Lemma 4.10 in (Fabert et al., 2016), and $E_\infty$ contains $C_c^\infty(\mathbb{R})$, which is dense in any Sobolev space. In this example, it is clear that points in higher scales have higher regularity and decay rates at infinity since the higher norms incorporate more derivatives and the exponential weights are increasing.

For two sc-Banach spaces $E$ and $F$ a map $\psi : E \to F$ is called *scale continuous*, or sc$^0$, if $\psi|_{E_k} : E_k \to F_k$ is continuous for all $k \in \mathbb{N}$. Furthermore, a sc-continuous map $\psi$ is called *scale differentiable*, or sc$^1$, if there exists a sc-continuous map $D\psi : E^1 \oplus E \to F$, such that $D\psi_\xi : E \to F$ is a bounded linear operator and for $\xi, h \in E_1$

$$\lim_{\|h\|_1 \to 0} \frac{\|\psi(\xi + h) - \psi(\xi) - D\psi_\xi(h)\|_0}{\|h\|_1} = 0. \tag{3}$$

Note that the base point $\xi$ and the step $h$ are always one scale higher than the image norm and $D\psi$ is only required to exist on $E_1$. This, together with the stronger convergence of $h$ in the $E_1$ norm, makes sc-differentiability a weaker condition than ordinary differentiability, assuming sc-continuity is already established. The tangent space of $E$ at a point $\xi \in E^1$ is defined as $T_\xi E := E$.

A map $\psi : E \to F$ is called sc$^k$, for $k \in \mathbb{N}$, if the above construction can be iterated $k$ times for the respective differentials and *scale smooth*, or sc$^\infty$, if it is sc$^k$ for any $k$.

In the theory of M-polyfolds, retracts play a similar role as open sets of $\mathbb{R}^n$ play for manifolds. A sc-smooth *retraction* on a sc-Banach space $E$ is a sc-smooth map $r : E \to E$ that satisfies the projection property $r \circ r = r$. The image of a sc-smooth retraction is called a sc-smooth *retract*. If $r$ is a retraction and $O := \operatorname{Im} r$ the associated retract, the tangent space of $O$ at $\xi \in O \cap E_1$ is defined as $\operatorname{Im} Dr_\xi$.

A *manifold-like polyfold*, or M-polyfold, is a paracompact Hausdorff space that admits an atlas with charts constructed out of retracts, giving rise to a rich structure in the same way as on manifolds. For the purposes of this paper, we are content with a single retract. This is analogous to a manifold that can be covered by a single chart.

**Retracts for $\Omega_1^1$**  We now show that the space $\Omega_1^1$ can be described by a single retract. Let $\tau_1, \tau_2 \in \mathbb{R}$, with $\tau_1 < \tau_2$, set $J := [\tau_1, \tau_2]$, and define $\beta : \mathbb{R} \setminus J \to \mathbb{R}$ as

$$\beta(\tau) = \exp\left(1/(\tau_1 - \tau) + 1/(\tau - \tau_2)\right).$$

Let $L^2(\mathbb{R})$ be the sc-Banach space with weighted Sobolev scales in (2), fix a $\gamma \in C_c^\infty(\mathbb{R}) \subset E_\infty$ with $\|\gamma\|_{L^2(\mathbb{R})} = 1$, define $\gamma_\tau(s) := \gamma(s + \beta(\tau))$, and abbreviate $\rho_\tau := d/d\tau\, \gamma_\tau$. Note that when $\tau$ goes to $\tau_1$, $\beta$ diverges, and since $\gamma_\tau$ is just a $\beta$-translate of $\gamma$, one essentially shifts the compact support of $\gamma_\tau$ to infinity as $\tau$ goes to $\tau_1$. Thus, since any function $f \in L^2(\mathbb{R})$ must decay at infinity, the $L^2$ inner product $\langle f, \gamma_\tau \rangle$ must tend to zero as $\tau$ goes to $\tau_1$ (or similarily $\tau_2$). This gives a mechanism for the dimensional collapse at $\tau_1, \tau_2$ which is encoded in our retraction defined below.

We define a *sc*-Banach space $E := \mathbb{R} \oplus \mathbb{R} \oplus L^2(\mathbb{R})$, denote a point in this space as $\xi = (\tau, x, f)$ with $\tau \in \mathbb{R}$, $x \in \mathbb{R}$, and $f \in L^2(\mathbb{R})$, and define the map $r : E \to E$,

$$r(\tau, x, f) = \begin{cases} (\tau, x, \langle f, \gamma_\tau \rangle \gamma_\tau), & \tau \in \mathbb{R} \setminus J \\ (\tau, x, 0), & \tau \in J \end{cases}. \tag{4}$$

For fixed $\tau$, the third component of $r$ is a linear projection, and it respects the scales of $E$, as $0, \gamma_\tau \in L^2(\mathbb{R})_\infty$. Thus $r$ is a retraction on $E$.

To see that $r$ is sc$^0$ we need only investigate $\langle f, \gamma_\tau \rangle \gamma_\tau$ at $\tau = \tau_1$ and $\tau = \tau_2$. Let $I_\tau$ be the support of $\gamma_\tau$. Assuming that $\tau \notin J$ is so close to $\tau_{1,2}$ that $\alpha(I_\tau) = 1$ we derive the following basic estimates from the definition of the scales.

$$\|\gamma_\tau\|_k \leq \sup_{s \in I_\tau} \exp(\delta_k s) \|\gamma\|_{W^{2,k}}$$

$$\leq C_1(\delta_k, \gamma) \exp(\delta_k \beta(\tau)),$$

and

$$\|f\|^2_{L^2(I_\tau)} \leq \mathrm{supp}_{s \in I_\tau} \exp(-2\delta_k s) \int_{I_\tau} |f(s)|^2 \exp(2\delta_k|s|)\, ds$$

$$\leq C_2(\delta_k, \gamma) \exp(-2\delta_k \beta(\tau)) \|f\|^2_{k, I_\tau}$$

Combining these, and using the Cauchy-Schwartz inequality for the $L^2$ inner products, yields

$$\|\langle f, \gamma_\tau \rangle \gamma_\tau \|_k \leq C \|f\|_{k, I_\tau}.$$

Take a sequence $(\tau_{1,i}, f_i)$ converging to $(\tau_1, f)$ in the k norm, $\tau_{1,i} < \tau_1$ for all $i \in \mathbb{N}$. Then we have $\|f_i\|_{k, I_{\tau_i}} \to 0$ as the compact set $I_{\tau_i}$ is shifted to infinity since $\beta$ has singularities at $\tau_1$ and $\tau_2$.

The same kind of estimates show that the scale differential of $r$ is given by the classical differential

$$Dr_{(\tau,x,f)}(\sigma, y, g) = \begin{cases} (\sigma, y, \sigma \langle f, \gamma_\tau \rangle \rho_\tau + \sigma \langle f, \rho_\tau \rangle \gamma_\tau + \langle g, \gamma_\tau \rangle \gamma_\tau), & \tau \in \mathbb{R} \setminus J \\ (\sigma, y, 0), & \tau \in J \end{cases}.$$

We check the candidate for $Dr$ given above at $\tau = \tau_1$ and $\sigma < 0$. In particular we will see that third component of $Dr$ tends to 0 in all scales at $\tau_1$. The case at $\tau = \tau_2$ is completely analogous. Let $(\sigma, y, g) \in E_1$, $(\tau_1, x, f) \in E_1$. First, a direct calculation shows that

$$\|r((\tau_1, x, f) + (\sigma, y, g)) - r((\tau_1, x, f) - Dr_{(\tau_1, x, f)}(\sigma, y, g))\|_0 = |\langle f + g, \gamma_{\tau_1 + \sigma} \rangle|.$$

We can then estimate the difference quotient (3) of $r$ directly

$$\lim_{\|(\sigma, y, g)\|_1 \to 0} \frac{|\langle f + g, \gamma_{\tau_1 + \sigma} \rangle|}{|\sigma| + |y| + \|g\|_1} \leq \lim_{\|(\sigma, y, g)\|_1 \to 0} \frac{C\|f + g\|_1 \exp(-\delta_1 \beta(\tau_1 + \sigma))}{|\sigma| + |y| + \|g\|_1}$$

$$\leq \lim_{\|(\sigma, y, g)\|_1 \to 0} \frac{C\|f + g\|_1 \exp(-\delta_1 \beta(\tau_1 + \sigma))}{|\sigma|}$$

$$= 0.$$

We need to check that $Dr$ is $sc^0$ at $\tau = \tau_1$ and $\tau = \tau_2$. Let $((\tau_i, x_i, f_i), (\sigma_i, y_i, g_i))$ be a sequence in $(E^1 \oplus E)_k$ converging to, say, $((\tau_1, x, f), (\sigma, y, g))$ with $\tau_i < \tau_1$ for all $i \in \mathbb{N}$. We only need to show that the third component of $Dr$ converges to 0.

$$\|Dr_{(\tau_i, x_i, f_i)}(\sigma_i, y_i, g_i)_3\|_k = \|\sigma_i \langle f_i, \gamma_{\tau_i} \rangle \rho_{\tau_i} + \sigma_i \langle f_i, \rho_{\tau_i} \rangle \gamma_{\tau_i} + \langle g_i, \gamma_{\tau_i} \rangle \gamma_{\tau_i}\|_k$$

$$\leq C|\sigma_i| \|f_i\|_{k+1} \frac{\exp(-(\delta_{k+1} - \delta_k)\beta(\tau_i))}{(\tau_i - \tau_1)^2} + C\|g_i\|_{k, I_{\tau_i}}$$

The first term goes to 0 since $\sigma_i$ and $f_i$ are bounded in the respective norms, $\delta_{k+1} > \delta_k$ and $\beta$ has a positive singularity at $\tau_1$. The second term goes to 0 since $g_i$ is bounded in the $k$ norm and the domain of integration $I_{\tau_i}$ is shifted to infinity. The same estimate also shows boundedness of the linear map $Dr_{(\tau, x, f)}$. Iterated calculations show that $r$ is $sc^\infty$, see (Hofer et al., 2010, Example 1.22) where a related construction is discussed.

Intuitively, the shifted $\gamma_\tau$, controlled by $\beta$, ensures that the third component of the retract $r$ collapses sufficiently fast for the limits in difference quotient (3) defining sc-differentiation to exist.

**Tangent space of $\Omega^1_1$**   Taking the $\tau$-derivative of $1 = \|\gamma_\tau\|^2_{L^2(\mathbb{R})}$ we see that $\gamma_\tau$ and $\rho_\tau$ are $L^2(\mathbb{R})$-orthogonal. This means that at $\tau \in \mathbb{R} \setminus J$, the tangent space of $O = \mathrm{Im}\, r$, with $r$ as in (4), is

$$T_{(\tau, x, f)}O = \mathrm{span}\{(1, 0, \langle f, \gamma_\tau \rangle \rho_\tau), (0, 1, 0), (0, 0, \gamma_\tau)\}.$$

The retract $O$ as a subspace of $E_0$, is homeomorphic to $\Omega^1_1 = (\mathbb{R} \setminus J \times \mathbb{R} \times \mathbb{R}) \cup (J \times \mathbb{R} \times \{0\})$, with the subspace topology of $\mathbb{R}^3$, via the map $\eta : O \to \Omega^1_1$

$$\eta(\tau, x, f) = \begin{cases} (\tau, x, \langle f, \gamma_\tau \rangle), & \tau \in \mathbb{R} \setminus J \\ (\tau, x, 0), & \tau \in J \end{cases}. \tag{5}$$

The differential of $\eta$ is given by

$$D\eta_{(\tau,x,f)}(\sigma, y, g) = (\sigma, y, \sigma\langle f, \rho_\tau\rangle + \langle g, \gamma_\tau\rangle)$$

Recalling that for $(\tau, x, f) \in O$ we have $f \in \mathrm{span}\{\gamma_\tau\}$ yields that the tangent space of $\Omega_1^1$ is simply $\mathbb{R}^3$,

$$T_{\eta(\tau,x,f)}\Omega_1^1 = D\eta_{(\tau,x,f)}(T_{(\tau,x,f)}O) = \mathrm{span}\left\{(1,0,0),(0,1,0),(0,0,1)\right\}.$$

This shows that the tangent spaces on $\Omega_1^1$ are the ones we expect from a subset of $\mathbb{R}^3$, even with the new polyfold structure, whereas the tangent spaces of the retract have a more involved structure. We will return to this in the next section, when we discuss vector fields and their flows.

The constructions of this section generalise in a straightforward manner. The second coordinate, denoted by $x \in \mathbb{R}$ above, can be replaced by a coordinate in $\mathbb{R}^n$ since the construction does not depend on it. More importantly, we may increase the jump in dimensions to $m$ by choosing an $L^2$-orthonormal family $\{\gamma_i\}_{i=1}^m$ of smooth compactly supported functions instead of a single $\gamma$. For $\tau \notin J$ the retraction $r$ is then given by $r(\tau, x, f) = (\tau, x, \pi_\tau(f))$, for $\tau \in \mathbb{R} \setminus J$, where $\pi_\tau$ is the $L^2$-projection to $\mathrm{span}\{(\gamma_i)_\tau\}_{i=1}^m$. In this way we can construct M-polyfolds $\Omega_n^m \cong (\mathbb{R} \setminus J \times \mathbb{R}^n \times \mathbb{R}^m) \cup (J \times \mathbb{R}^n \times \{0\})$ for arbitrary $n, m \in \mathbb{N}$.

## 2.2 Flows and ODEs on M-polyfolds

To extend NODE models to M-polyfolds, we need vector fields that are compatible with the sc-smooth structure. A $\mathrm{sc}^k$ vector field on a sc-Banach space $E$ is a $\mathrm{sc}^k$ map $X : E^1 \to E$, and we denote the space of such vector fields on $E$ by $\mathfrak{X}^k(E)$. On a retract $O \subset E$, with retraction $r$, we can view the set of vector fields as $\mathfrak{X}^k(O) = \{X \in \mathfrak{X}^k(E) \mid X_\xi \in T_\xi O, \forall \xi \in O \cap E^1\}$.

To construct the flow generated by a vector field $X$, we need to solve ODEs on $E$. If a vector field $X$ is actually in $\mathrm{sc}^0(E^1, E^1)$ then, since $E$ is a filtration of Banach spaces, ODE solution theory applies independently on every scale $E_k$. Thus, ODEs on $E$ can be solved on every scale separately. If additionally $X$ is Lipschitz continuous on every scale, standard uniqueness arguments imply the solutions for different scales agree when restricted to the same scale. This generalises to the theory of flows by similar arguments for the dependence on the initial conditions. Given $X \in \mathfrak{X}^k(O)$ on a retract $O$ with retraction $r$, we can consider a lift, i.e. $\tilde{X} \in \mathfrak{X}^k(E)$ such that $Dr_\xi\tilde{X}(\xi) = X(r(\xi))$. If $\tilde{X}$ can be integrated to a flow $\tilde{\phi}$ on $E^1$, then it gives rise to a flow $r \circ \tilde{\phi}$ on $O \cap E^1$. We discuss ODEs and flows on sc-Banach spaces and M-polyfolds in greater detail in a forthcoming article. For ODEs on Banach spaces, see, for instance, Chapter 16 in (Pata, 2019).

However, in the case of the retraction (4) for the $\Omega_1^1$ model, this construction is not applicable, since at $\tau_1$ and $\tau_2$ any lift of a vector field whose flow traverses the bottlenecks will be singular as a map from $E^1$ to $E^1$. This is expected because at these points, any flow entering the dimensional bottleneck must lose injectivity and thus cease to be a flow. This is a fundamental property of stratified spaces like $\Omega_1^1$ and indeed the main feature we use for the autoencoder construction. Thus, what we are really interested in for the autoencoder application is (forward) semi-flows that traverse the dimensional bottleneck. That is maps $\phi(\cdot, \xi) : \mathbb{R}_+ \to O$ such that for all $t, s \in \mathbb{R}_+$ we have $\phi(t + s, \xi) = \phi(t, \phi(s, \xi))$ and which solve the flow ODE.

A semi-flow $\phi$ that is $\mathrm{sc}^0$ at $\tau_1$ and $\tau_2$ has a component $\phi_3$ which converges to 0 at these points in any scale by definition of scale continuity. In coordinate charts $\eta$ (5), this corresponds to super-exponential decay. For example, suppose $\lim_{t\to t_1} \phi(t)_1 = \tau_1$ from below, then, from the estimate $\tilde{C}\exp(\delta_k\beta(\tau)) \leq \|\gamma_\tau\|_k \leq C\exp(\delta_k\beta(\tau))$ and since we can write $\phi(t)_3 = \eta(\phi(\tau))_3\gamma_{\phi(t)_1}$, we get the requirement

$$\lim_{t\to t_1} \eta(\phi(t))_3 \exp\left(\delta_k\beta(\phi(t)_1)\right) = 0 \quad \forall k \in \mathbb{N}. \tag{6}$$

Throughout, subscripts are used to indicate specific elements of our vectors. This holds both for points in the embedding and on the retract.

In general, vector fields on the retract $O = \mathrm{Im}\, r$ have the form

$$X_{(\tau, x, \langle f, \gamma_\tau\rangle\gamma_\tau)} = (X_1, X_2, X_1\langle f, \gamma_\tau\rangle\rho_\tau + X_3\gamma_\tau),$$

for functions $X_i : O \to \mathbb{R}$, $i \in \{1, 2, 3\}$. This means that for $\tau \in \mathbb{R} \setminus J$ we can view vector fields on $\Omega_1^1 \subset \mathbb{R}^3$ as classical vector fields $Y_{(\tau, x, y)} = (Y_1, Y_2, Y_3)$, which lift to $\mathrm{sc}^0$ vector fields

$$X_{(\tau, x, f)} = D\eta^{-1}(Y(\eta(\tau, x, f))) = (Y_1, Y_2, Y_1 y \rho_\tau + Y_3 \gamma_\tau) \, . \tag{7}$$

For the third component the condition for $X$ above to be $\mathrm{sc}^0$ amounts to

$$\lim_{\tau \to \tau_i} Y_3(\tau, x, \langle f, \gamma_\tau \rangle) \exp(\delta_k \beta(\tau)) = 0 \quad \forall k \in \mathbb{N}, \quad i \in \{1, 2\}. \tag{8}$$

Using techniques like separation of variables, it is straightforward to find vector fields $Y = (Y_1, Y_2, Y_3)$ whose semi-flows satisfy (6). Note that condition (6) is fulfilled if there is a $t_0 < t_1$ such that $\eta(\phi(t))_3 = 0$ for all $t \in [t_0, t_1]$. In Section 3.1, we give an example of a vector field with an associated semi-flow with this property.

With slight abuse of notation, we let $\phi$ denote both the semi-flow on $\Omega_1^1$ generated by the vector field $Y$ and its image under $\eta^{-1}$. See Figure 1b) for an illustration of a semi-flow in $\Omega_1^1$. The generalisation to vector fields and semi-flows on $\Omega_n^m$ is straightforward, as indicated in Section 2.1.

## 2.3 PolyNODEs: Neural ODEs on M-polyfolds

Having defined the necessary machinery from polyfold theory, we can now extend the definition of a neural ODE to stratified spaces. Let $E$ be a sc-Banach space, $r : E \to E$ a retraction, and $O = \mathrm{Im}\, r$ the corresponding retract. A *PolyNODE* parametrised by the vector field $X \in \mathfrak{X}^k(O)$ is the map $h : O \to O$, $h(\xi) = \phi(1, \xi)$, where $\phi : \mathbb{R}_+ \times O \to O$ is the semi-flow generated by $X$,

$$\frac{d}{dt}\phi(t, \xi) = X(\phi(t, \xi)), \quad \phi(0, \xi) = \xi \, . \tag{9}$$

Since forward integration of the semi-flow is well-defined, the map $h$ is also well-defined. However, it is not invertible due to the lack of injectivity of $\phi$. The generalisation to M-polyfolds covered by several retracts is also straightforward, but not required for the case $\Omega_n^m$ we consider below.

The semi-flow $\phi$ generated by $X$ can traverse the stratification points at the intersection of the regular components of the underlying space, meaning that the dimension of the state vector $\phi(t, \xi)$ can change in a sc-differentiable way. This construction of PolyNODEs is our first major contribution; a flow-based machine learning model capable of accommodating variable-dimension dynamics.

The $\Omega_n^m$ spaces described in detail above can be easily generalised to have multiple successive intrinsic dimension changes by adding singularities to the shift function $\beta$. In this way, PolyNODEs corresponding to more general neural network architectures beyond autoencoders can be constructed.

# 3 A PolyNODE Autoencoder Model on $\Omega_n^m$

Having defined PolyNODEs in the previous section, we proceed to construct explicit examples of such models. The purpose is to demonstrate the viability of PolyNODEs in machine learning by showing that M-polyfolds corresponding to specific machine learning problems can be constructed and that these spaces admit vector fields which probe their M-polyfold structure. In the subsequent section, we then present numerical experiments that further demonstrate that *sc*-NODE models defined by these vector fields can be trained to solve the machine learning tasks.

## 3.1 Autoencoder Geometry and Data Embedding

As a proof-of-concept, we consider the problem of reconstructing geometric objects using a PolyNODE with a dimensional bottleneck similar to that of a traditional autoencoder, for which the appropriate geometry is the space $\Omega_n^m$ with $J = [\tau_1, \tau_2]$ described in detail above. We can then interpret the region $\tau < \tau_1$ as the input space, the stratification at $\tau = \tau_1$ as the encoding, $\tau \in (\tau_1, \tau_2)$ as the latent space, the stratification at $\tau = \tau_2$ as decoding, and $\tau > \tau_2$ as the output space of the autoencoder structure; see Figure 2 for an illustration.

For an index set $I$, we sample a submanifold $S \subset \mathbb{R}^{1+n+m}$ of dimension $\dim S \leq n+1$, $\{z_i\}_{i \in I}$, $z_i \in S$. We choose two embeddings $\iota_1$ for the input data and $\iota_2$ for the target data into $\Omega_n^m$, such that $\iota_1(z_i)_1 < \tau_1$ and $\iota_2(z_i)_1 > \tau_2$ for all $i \in I$. An easy way to achieve this is to embed the samples $z_i \in S \subset \mathbb{R}^{n+m}$ into $\Omega_n^m$ as $\mathcal{X} = \{(\tau_{\mathcal{X}}, z_i)\}_{i \in I}$ for a fixed $\tau_{\mathcal{X}} < \tau_1$ to obtain the input data and $\mathcal{Y} = \{(\tau_{\mathcal{Y}}, z_i)\}_{i \in I}$ for some fixed $\tau_{\mathcal{Y}} > \tau_2$ to obtain the target data. We use this type of embedding in our main experiment, the autoencoding of the spiral, and assume it in the following.

We restrict ourselves to vector fields whose semi-flows actually traverse the bottleneck. Furthermore, we regard the $\tau$ coordinate in $\Omega_n^m$ as artificial, corresponding to the depth dimension in a traditional autoencoder, and consequently prescribe a constant velocity in this direction by enforcing $X_1 = \tau_{\mathcal{Y}} - \tau_{\mathcal{X}}$. A latent representation of a point $p \in \mathcal{X}$ is given by $\phi(t, p)$ for any $t$ such that $\phi(t, p)_1 \in [\tau_1, \tau_2]$. We may always choose $\Omega_n^m$ and the embeddings of the data such that this is realised at $t = 1/2$.

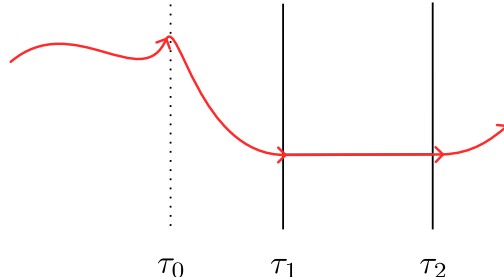

$$\tau_0 \qquad \tau_1 \qquad \tau_2$$

Figure 2: Illustration of flow line entering the bottleneck at $\tau_1$ and exiting at $\tau_2$, with compression onset at $\tau_0$.

## 3.2 Parameterisation of Compressing Vector Fields

In order to construct vector fields $X$ for the PolyNODE model that can accomplish the dimensional reduction in the encoding phase, we need to ensure convergence of the semi-flow to the stratification point $\tau = \tau_1$ in finite time. This is accomplished by choosing the parametrisation in (7) and imposing conditions on the components $Y_{1+n+j}$, $j = 1, \ldots, m$, compressing the semi-flow in the directions transverse to the latent space.

Let $\tau_0 < \tau_1$ define the compression region, let $k_j \in C^{0,1}(\mathbb{R}^{1+n+m}, \mathbb{R}_+)$, with $0 < K_j < k_j$ for constants $K_j$, and let $a \in (0, 1)$. For $p \in (\tau_0, \tau_1) \times \mathbb{R}^{n+m}$ we then restrict to vector fields

$$Y_{1+n+j}(p) = -k_j(p) \operatorname{sign}(y_j) |y_j|^a, \tag{10}$$

which guarantees convergence of the semi-flow to $y_j = 0$ in finite time (Bhat & Bernstein, 2000). Specifically, for a constant rate function $k_j(p) = K_j$ the time to convergence is given by $T(y_j^0) = |y_j^0|^{1-a}/(K_j(1-a))$, where $y_j^0 \neq 0$ is the initial value. This means, by choosing the lower rate bounds $K_i$ appropriately, we can guarantee that the semi-flow $\phi$ maps a bounded subset of the region $[\tau_0, \tau_1) \times \mathbb{R}^{n+m}$, containing the slice at $\tau = \tau_0$ of all flow lines originating on the input data $\mathcal{X}$, to $(\tau_0, \tau_1) \times \mathbb{R}^n \times \{0\}$ before it arrives at the stratification $\tau = \tau_1$. Thus $\phi$ restricted to that bounded set satisfies condition (6) and is therefore an $sc^0$ semi-flow. The vector field components in (10) lift to an $sc^0$ vector field, that is they satisfy estimate (8), since we may choose the sequence $\{\delta\}_{k \in \mathbb{N}}$ such that $a\delta_{k+1} > \delta_k$.

In this way, we obtain a family of $sc^0$ compressing vector fields by parametrising the functions $k_j$ using neural networks. The components $Y_{1+i}$, $i = 1, \ldots, n$ corresponding to the $x_i$ directions parallel to the latent space are unrestricted and parametrised directly by neural networks. The same is true for $Y_{1+n+j}$ for $\tau < \tau_0$ and $\tau > \tau_2$.

Note that away from $y_j = 0$, the $Y_{1+n+j}$ in (10) are smooth, but at $y_j = 0$ they are only Hölder continuous. ODE theory still guarantees the existence of a semi-flow, which is no longer injective after reaching $y_j = 0$; in particular, flow lines may merge. Furthermore, note that the general PolyNODE theory allows for $sc^k$ vector fields of higher regularity than the particular family of compressing vector fields we describe here and use in the numerical experiments below.

### 3.3 Numerical Implementation and Backpropagation

All compressing vector fields are inherently non-Lipschitz with singular derivatives. Thus, standard gradient-based optimisation requires an additional numerical regularisation, since the backward pass involves differentiating the vector field. For our choice of vector field (10), the singularity are of the form $|y_i|^{a-1}$ at $y_i = 0$. We regularize by introducing a cut-off for the vector field for small values of $y_i$.

For simplicity, we use a cut-off function $\varphi$ based on a polynomial,

$$\varphi(x) = \begin{cases} 0 & \text{for } x < 0 \\ (3 - 2x)x^2 & \text{for } 0 \le x \le 1 \\ 1 & \text{for } x > 1 \end{cases}, \quad \varphi'(x) = \begin{cases} 0 & \text{for } x < 0 \\ 6(1 - x)x & \text{for } 0 \le x \le 1 \\ 0 & \text{for } x > 1 \end{cases}.$$

Scaling $\varphi$ gives a cut-off function $\varphi_{b,c} := \varphi((x - b)/(c - b))$ in the interval $(b, c)$, $b \ge 0$. In the forward pass, the vector field then reads

$$Y_{1+n+j}(p) = -k_j(p)\,\text{sign}(y_j)|y_j|^a \varphi_{b,c}(|y_j|).$$

This introduces numerical errors in the compressed latent coordinates $y_j$ of at most order $b$. The parameters $b$ and $c$ therefore allow us to explicitly control the trade-off between regularity and compression; see Appendix B. Although the cut-off raises the regularity of the vector field from $\text{sc}^0$ to classically smooth, it acts only in a small neighbourhood of the singularity and should therefore be regarded as a numerical device rather than a modification of the dynamics.

We then implement the corresponding regularised derivative, which is used in the custom backward method of the vector field. In the backwards pass, the derivative is given as

$$\nabla Y_{1+n+j}(p) = -\nabla k_j(p)\,\text{sign}(y_j)|y_j|^a \varphi_{b,c}(|y_j|) - k_j(p)a|y_j|^{a-1}\varphi_{b,c}(|y_j|)e_j - k_j(p)|y_j|^a \frac{\varphi'_{b,c}(|y_j|)}{c - b}e_j,$$

where $\{e_j\}$ is the standard Euclidean basis. This derivative does not have a singularity at 0. This modified backpropagation allows us to use the adjoint sensitivity method in (Chen, 2018).

In principle, any smooth cut-off function could be used. However, the polynomial cut-off is particularly convenient because the singularity term $|y_j|^{a-1}$ in the derivative of the vector field is cancelled algebraically by the polynomial factors of the cut-off. For another choice of compressing vector field, another choice of cut-off function might be better suited.

## 4 Experiments: Reconstruction of Geometrical Objects

In this section, we conduct several numerical experiments based on the PolyNODE autoencoder model constructed above[1]. All experiments are based on parametrised families of the compressing vector fields in (10) with a reduced setup detailed in Table 1, in particular we have fixed the compression rates $k_j = 25$, $a = 1/2$, and we have no latent dynamics in the region $[\tau_1, \tau_2]$. Recall that we keep the speed of $Y_1$ constant to ensure a total time $T = 1$ and a latent time of $1/2$.

Further experimental details are provided in Appendix B. The implementation in PyTorch is found in (Authors, 2026) at [omitted for reviewing]. In all experiments described below, we train our PolyNODEs using the adjoint sensitivity method (Chen et al., 2018) with the custom implementation of the backward pass for the compressing vector fields in Section 3.3.

### 4.1 Reconstruction of Spirals

Our goal is to autoencode a family of spirals, where the number of turns ranges from $N = 1/2$ to $N = 5$, and extract a latent representation. The spiral is one dimensional and naturally embeds in two dimensions. We

---

[1]In this section, we use the term flow to refer to all collections of integer curves that appear, dropping the explicit semi-flow qualification.

Table 1: Structure of vector fields for reconstruction experiments.

| Region | Shape of Vector Field | Neural Networks | Input |
|---|---|---|---|
| $\{\tau < \tau_0\}$ | $(C, Y_2, ..., Y_{n+m+1})$ | $\{Y\}_2^{n+m+1}$ | $(\tau, x, y)$ |
| $\{\tau_0 < \tau < \tau_1\}$ | $(C, 0, ..., 0, Y_{n+2}, ...Y_{n+m+1})$, $\{Y_i\}_{i=n+2}^{n+m+1}$ as in (10), | - | $(\tau, x, y)$ |
| $\{\tau_1 < \tau < \tau_2\}$ | $(C, 0, ..., 0)$ | - | - |
| $\{\tau > \tau_2\}$ | $(C, Y_2, ..., Y_{n+m+1})$ | $\{Y\}_2^{n+m+1}$ | $(\tau, x, y)$ |

add an additional compressing dimension to allow for more flexible flow-lines. Consequently, we choose $\Omega_1^2$ with $\tau_1 = 0$ and $\tau_2 = 1$ as an M-polyfold, and $\tau_0 = -3$ for the onset of the compressing vector fields. For the embedding we choose $\tau_{\mathcal{X}} = -7$ and $\tau_{\mathcal{Y}} = 8$, hence the speed is $C = 15$. For samples $\{z_i\}$, we obtain the input data set $\mathcal{X} = \{p_i\}_{i \in I} = \{(\tau_{\mathcal{X}}, z_i)\}_{i \in I}$, and the target dataset $\mathcal{Y} = \{(\tau_{\mathcal{Y}}, z_i)\}_{i \in I}$.

For the training of the model, we employ a loss function $L = L_1 + \lambda(L_2 + L_3)$ with three loss terms, with weight $\lambda = 20$ for the last two, where

$$L_1 = \frac{1}{|I|} \sum_{i \in I} |(\phi(1, p_i) - (\tau_{\mathcal{Y}}, z_i))|^2,$$

$$L_2 = \frac{1}{|I|} \sum_{i \in I} |(\phi(t_0, p_i)_3, \phi(t_0, p_i)_4) - (1, 1)|^2,$$

$$L_3 = \frac{1}{|I|} \sum_{i \in I} \left| d_S(p_i, q_{p_i}) - c_d |\phi(t_0, p_i)_2 - \phi(t_0, q_{p_i})_2| \right|.$$

Here, $c_d$ is the square root of the intrinsic diameter of the spiral, chosen this way since the intrinsic distance scales with the square of the angular parameter, and $d_S$ is the distance function on the spiral. Additionally, $q_p$ is a random element of $\mathcal{X}$, such that $q_p \neq p$, $\forall p \in \mathcal{X}$ and $q_p \neq q_{p'}$ if $p \neq p'$.

The term $L_1$ is the mean squared error of the reconstructed spiral. The second term $L_2$ forces the flow to explore the fourth dimension and ensures that the $y_1$ and $y_2$ coordinates are in a range to be mapped to 0 by the compressing vector fields by the time the flow reaches $\tau = 0$. The last term $L_3$ enforces the flow to be approximately isometric at time $t_0 = 1/4$, which aids in the unwinding of the spiral to a line. Here $\tau_{\mathcal{X}}$ and $\tau_{\mathcal{Y}}$ are chosen such that at time $t_0$ the flow is at $\tau = \tau_0$, i.e. the onset of the compressing vector fields. For the spiral, we know the distance function $d_S$ explicitly. In general, the approximate distance function may be reconstructed from the data, see for instance (Mémoli & Sapiro, 2005). This last loss term is rather strong and problem-specific; however, it does not contain the entire information of the spiral parametrisation. Moreover, we stress that the goal of this example is to show that a vector field for an unwinding and autoencoding flow can be learned. In the next example, we show that a round sphere can be autoencoded using only the mean squared reconstruction error. Figure 3 illustrates the unwinding and encoding of the spirals to the line as well as the subsequent reconstruction. Note that the flow for $\tau < \tau_1$ is 4 dimensional, thus allowing for apparent crossing of flow lines.

The reconstruction error for spirals of different numbers $N$ of turns is shown in Figure 4, and the projected reconstruction of the spirals with $N = 2, 3, 4, 5$ is shown in Figure 5. The reconstruction degrades slightly with increasing number of turns, as expected, but remains below 1.2% (validation error) in all our experiments.

To determine how well the spirals are encoded onto the line, we first illustrate time slices of the unwinding in Figure 6. We then quantify the encoding by investigating the monotonicity of the map $s \mapsto \phi(1/2, f(s))_2$, where $f$ is the parametrisation of the spiral and $s \in [0, 2\pi]$. Figure 7 shows the monotonicity error relative to the number of sample points during training for spirals with $N = 2, 3, 4, 5$. The y-axis with the error is represented in a log scale to resolve the dynamics for low errors. At the start of the training, a significant fraction of the points are misaligned for large values of $N$, but as the training progresses the number of misaligned points tends to zero. In some cases, training results in perfect alignment, shown by omission of points in the log-scaled plot. In these cases, the spiral is perfectly encoded onto the line within the sample accuracy of 5000 equidistant points. All experiments show very good monotonicity on the sampled data

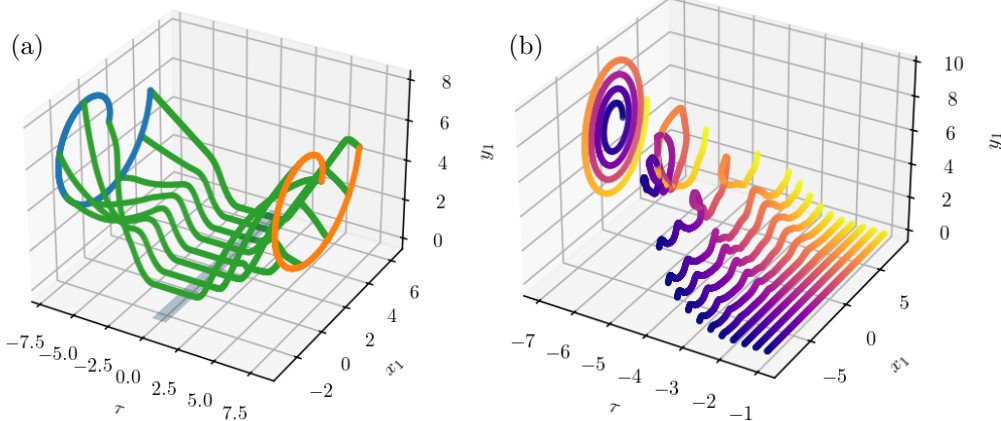

Figure 3: (a) Flow lines (green) for individual input samples for $N = 1$. Input set $\mathcal{X}$ (blue) and reconstructed output $\hat{\mathcal{Y}}$ (orange). The $y_2$ component is projected out for visualisation. (b) Time slices of the flow for a spiral with $N = 4$. Colour scale corresponds to the angular parameter.

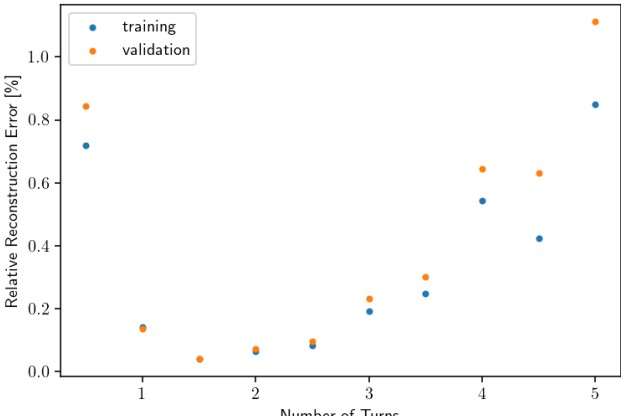

Figure 4: Mean Euclidean reconstruction error relative to number of spiral turns.

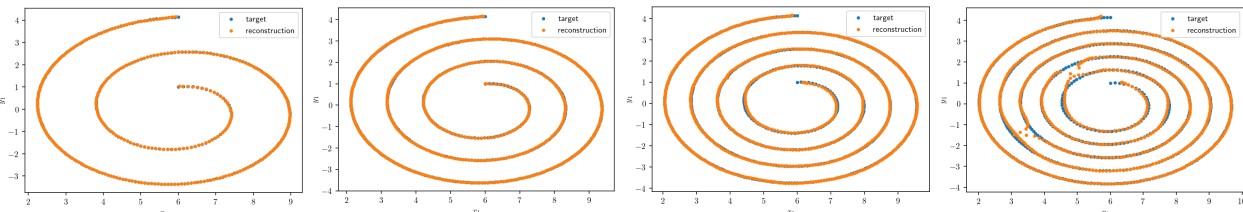

Figure 5: Reconstruction (2d projection) of spirals with number of turns $N = 2, 3, 4, 5$.

points, with fewer than $0.02\%$ misaligned points, even when the reconstruction starts to suffer slightly above $N = 3$. This is likely due to the rather strong approximate isometry loss $L_3$ for the unwinding and the explicit compressing vector field. Hence, the encoder and decoder are not symmetric.

In order to verify that the numerical results are independent of the random initialisation of the network layers, discussed further in Appendix B, we repeat the $N = 3$ spiral reconstruction for five (sequential integer) seeds. The monotonicity error during training in these runs is shown in Figure 8, confirming consistent behaviour of the monotonicity. Similarly, the reconstruction error at the end of training is consistent across different seeds.

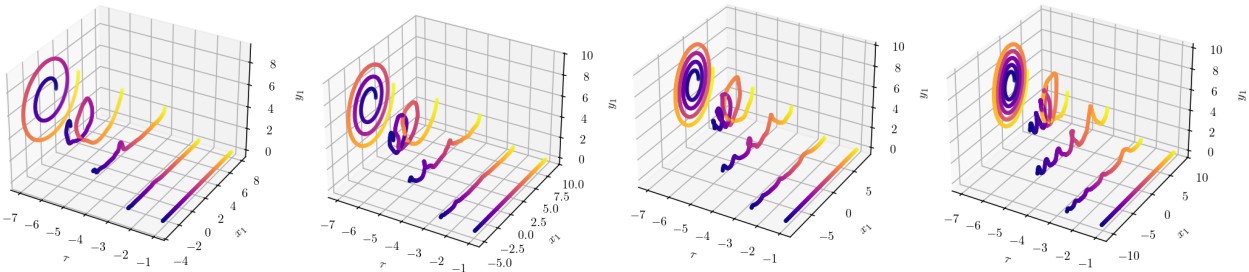

Figure 6: Unwinding and encoding of spirals with number of turns $N = 2, 3, 4, 5$.

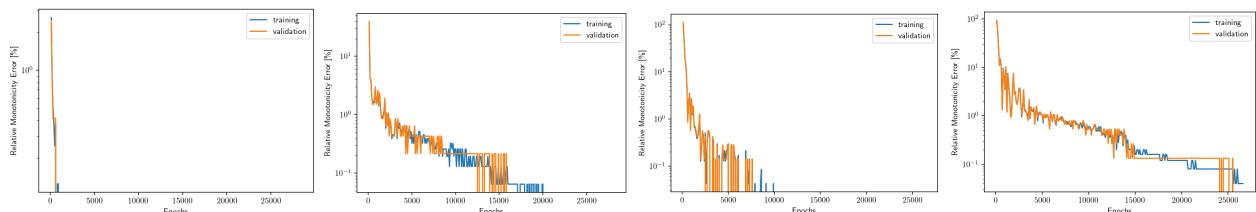

Figure 7: Monotonicity error of spirals with number of turns $N = 2, 3, 4, 5$.

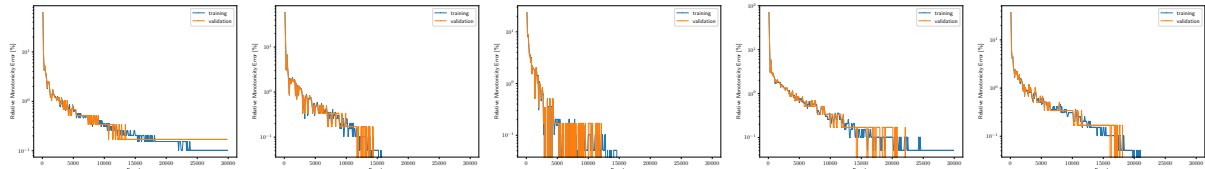

Figure 8: Monotonicity error for the spiral with $N = 3$ turns over 5 different seeds.

The monotonicity error in Figure 7 indicates that the state obtained after compression indeed captures information about the encoded object. This aspect is further explored by additional experiments in Appendix A, where a latent representation is extracted and used to classify points on the spirals.

## 4.2 Reconstruction of the Sphere

As a further example, consider the two-dimensional round sphere of radius 1 in $\mathbb{R}^4$ around a point $x_0$, given by $\{(z_1, z_2, 0, z_3) \mid z \in S_1^2(x_0) \subset \mathbb{R}^3\}$. The goal is to embed it into the three dimensional latent region $[\tau_1, \tau_2] \times \mathbb{R}^2$.

We choose $\Omega_2^1$ with $\tau_1 = 0$ and $\tau_2 = 3$ as an M-polyfold with $\tau_0 = -3$ for the onset of the compressing vector fields. We sample the sphere and, with $e_1 = (1, 0, 0, 0)$, embedded the samples as $\mathcal{X} = \{p_i\}_{i \in I} = \{z_i + \tau_{\mathcal{X}} e_1\}_{i \in I}$ and $\mathcal{Y} = \{z_i + \tau_{\mathcal{Y}} e_1\}_{i \in I}$. We choose $\tau_{\mathcal{X}} = -7$ as well as $\tau_{\mathcal{Y}} = 10$. This leaves us with a speed of $C = 17$. Here, we break the embedding convention of Section 3.1 for an easier visualisation.

The loss function is the mean squared reconstruction error,

$$L = \frac{1}{I} \sum_{i \in I} |(\phi(1, p_i) - (z_i + \tau_{\mathcal{Y}} e_1))|^2.$$

Figure 9 shows the encoding and decoding of the 2-sphere embedded in $\mathbb{R}^4$, via flow time slices. The third dimension (second latent dimension) is indicated by colour. Note that due to the data embedding, and contrary to the spiral example, flow time slices to do not coincide with $\tau$ slices any more. We see that, starting from the monochrome sphere where the third coordinate is zero, the flow explores the extra dimension while

the height (fourth dimension) is compressed. At the end of the encoding, the sphere is contained in $\mathbb{R}^3$. At 0.06%, the relative mean reconstruction error of the decoded sphere is very low.

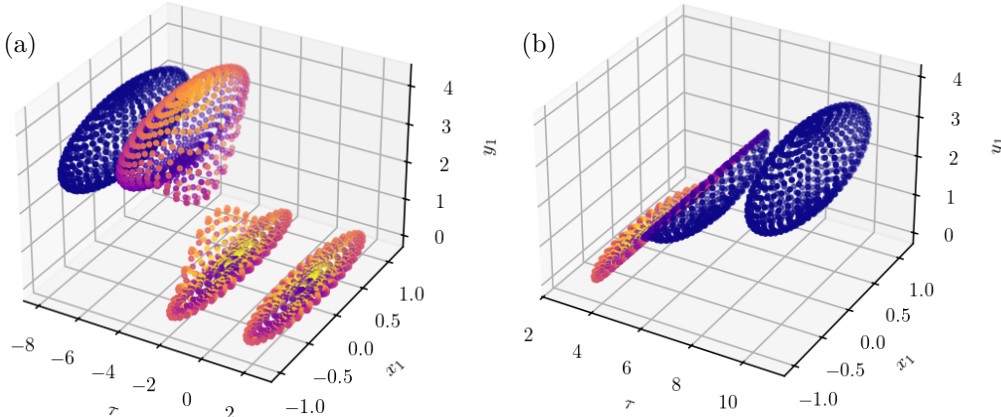

Figure 9: Flow time slices of the encoding (a) and decoding (b) of the 2-sphere in $\mathbb{R}^4$. The colour represents the coordinate in the third dimension (second latent dimension), with yellow being further from dark blue representing 0.

The experiments presented in this section demonstrate that variable-dimension PolyNODEs can be trained to solve reconstruction tasks using the standard backpropagation paradigm for NODEs, suitably adapted for flows on M-polyfolds. We emphasise that the experimental results should be interpreted as supporting this non-trivial result, not as a claim of superior performance on the specific tasks. Indeed, there are many standard machine learning models that would easily solve the reconstruction task, but none that would do so using flows to accomplish dimensional reduction in the encoding.

## 5 Conclusion

In this work, we demonstrate that it is possible to extend flow-based NODE models to M-polyfolds to accommodate variable-dimensional dynamics. The construction hinges on the use of infinite-dimensional Banach spaces as the underlying spaces where the flow dynamics plays out, even when the tangent spaces of the retracts and polyfolds we consider are finite-dimensional. The reason is that the non-trivial scale structures required to relax the differentiability condition and accomplish dimensional jumps cannot be defined in Euclidean spaces.

We believe there are several interesting consequences of the construction of our PolyNODE models. The generality and flexibility of NODE models are increased by allowing varying dimensions. We use the encoder-decoder structure as an example, but more generally, PolyNODEs can be defined to represent the continuum limit of feed-forward networks with arbitrary layer widths. Similar architectural flexibility is often used in practice to design accurate and efficient neural networks. Somewhat conversely, the PolyNODEs could also be leveraged to gain a better understanding of well-posedness, stability, and convergence for general architectures using ODE theory (cf. (Haber & Ruthotto, 2018; Thorpe & van Gennip, 2023)).

We construct an M-polyfold mirroring an autoencoder setup, together with a family of parametrised vector fields that traverse the dimensional bottleneck. Although we use problem-specific loss functions in the reconstruction tasks, the $\Omega_n^m$ space and the vector fields are problem-agnostic. Our experiments with this class of models demonstrate the ability to train PolyNODE models and extract meaningful latent representations in an autoencoder setting, but contain no quantitative evaluation on more realistic problems or comparisons with other model architectures. Nor do we explore the vast range of possible geometries and architectures that could be constructed from flows on M-polyfolds. Both these aspects indicate interesting future directions of research.

While using M-polyfolds to describe variable-dimensional NODE dynamics is mathematically principled, it has several disadvantages compared to other autoencoder architectures. First, the PolyNODE framework suffers from the same performance issues as ordinary NODEs since it uses the same solvers, exacerbated by large gradients during the compression (even after regularisation). Compared to the other NODE-based models discussed in the introduction, we therefore have higher numerical complexity, albeit with explicit control over the trade-off between compression and gradient stability. The standard NODE-based methods are also simpler, in that they require no involved construction of singular spaces or regularisations of the backpropagation, but for the simple geometric reconstruction tasks, we don't expect PolyNODEs to have an advantage over the standard models in terms of numerical reconstruction performance.

Even though the use of standard gradient-based training algorithms requires regularisation of the vector fields, we emphasise that the cut-off functions are highly spatially localised. Thus, the regularised vector fields still describe the appropriate exponential compression dictated by the M-polyfold structure away from the singular stratification loci. An interesting future line of research would be to remove the need for regularisation by developing polyfold native training algorithms using only the weaker sc-calculus notion of gradients.

As discussed in the introduction, the stratified spaces we consider in this paper can always be embedded in Euclidean space $\Omega_n^m \subset \mathbb{R}^{1+n+m}$. An important question is whether the construction of PolyNODEs enables us to learn dynamics – relevant for machine learning – which cannot be described in terms of flows in the ambient space. In a forthcoming publication, we address this question by constructing examples of flows that traverse the bottleneck in $\Omega_n^m$ but cannot be extended to ambient space. The dynamics generated by such vector fields are unique to the M-polyfold and cannot be represented in any ambient Euclidean space. We also develop a general theory of (semi-)flows generated by sc-smooth vector fields on M-polyfolds, extending the analysis carried out for the $\Omega_n^m$ spaces in this paper. We hope that this will establish the foundation for a new, exciting research direction in variable-dimension flow-based machine learning.

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

# A  Classification Experiments

Encoding–decoding architectures are often employed to enable downstream tasks based on the latent representation. We illustrate this capability using the latent representation learned in the spiral experiment, Section 4.1, to solve radial and angular classification tasks. In both cases, we define three labels $\{l_1, l_2, l_3\}$, described in detail in the following paragraphs, and extract latent states at $t = 1/2$ to produce labelled samples for the classifications.

In principle, any model can be used for the classification task, but we opt to stay in the NODE framework. We designate target points in latent space for each label and learn a vector field whose flow maps sample points to their corresponding target points.

The latent representation of the spiral is a one-dimensional curve lying in a two-dimensional plane. For a general ordering of labels along the latent curve, we cannot guarantee the existence of a vector field making points flow to their target point if the integral curves are restricted to lie in the plane. This is because crossings of integral curves would be necessary for all but the simplest monotone orderings. Thus, we augment the NODE to three dimensions.

The loss used is MSE between the last points on the integral curves and their corresponding target points. In addition, an attractor term for each target point is added to the vector field to promote flow towards a target point, giving the NODE

$$\frac{dx(t)}{dt} = Y_\theta(x, t) + C \sum_i \frac{y_i - x(t)}{|y_i - x(t)|} e^{-k(y_i - x(t))^2}, \tag{11}$$

where $y_i$ indicates the target point for label $l_i$ and $Y_\theta(x, t)$ is the learnable vector field. We choose $k$ so that there is a clear separation between the target point attractors. For details on the experimental setup and the neural networks used see Appendix B.2.

With sufficiently many parameters and an expressive enough architecture, any desirable accuracy is achievable for both classification problems. Therefore, the accuracy presented below should not be interpreted as an indication of superior model structure. Rather, it demonstrates the ability of PolyNODEs to extract latent representations of sufficient accuracy for downstream tasks.

## A.1  Radial Classification

We divide the interval $[0, R_{\max})$ into three equal parts, where $R_{max}$ is the maximal radius of the spiral. Each point in the latent space corresponds to a point on the spiral. A point in the latent space is given the label $l_i$, $i \in \{1, 2, 3\}$ if its corresponding point on the spiral has radius in $[R_{max}(i-1)/3, R_{max}i/3)$.

Due to the unwinding nature of the flow for the spiral autoencoder, the labelling in latent space becomes monotone for the radial classification problem. Consequently, a simple flow is expected and experimentally observed, see Figure 10a. The model reaches a peak accuracy of 100% in this task, and never drops below 80% accuracy after a single epoch.

## A.2  Angular Classification

Classes are set by the angle in the plane of the spiral. The interval $[0, 2\pi)$ is equally divided into $[2\pi(i-1)/3, 2\pi i/3)$, $i \in \{1, 2, 3\}$ and a latent point is assigned a label $l_i$ according to its corresponding point on the spiral.

For angular classification, the unwinding nature of the autoencoder flow causes the labels to switch several times along the latent line. Therefore, no simple classification flow is possible, and the augmentation is necessary. The experiment is qualitatively consistent with this expectation, see Figure 10b. The model reaches a maximum accuracy of 98% in this task and the accuracy never drops below 80% accuracy after 150 epochs.

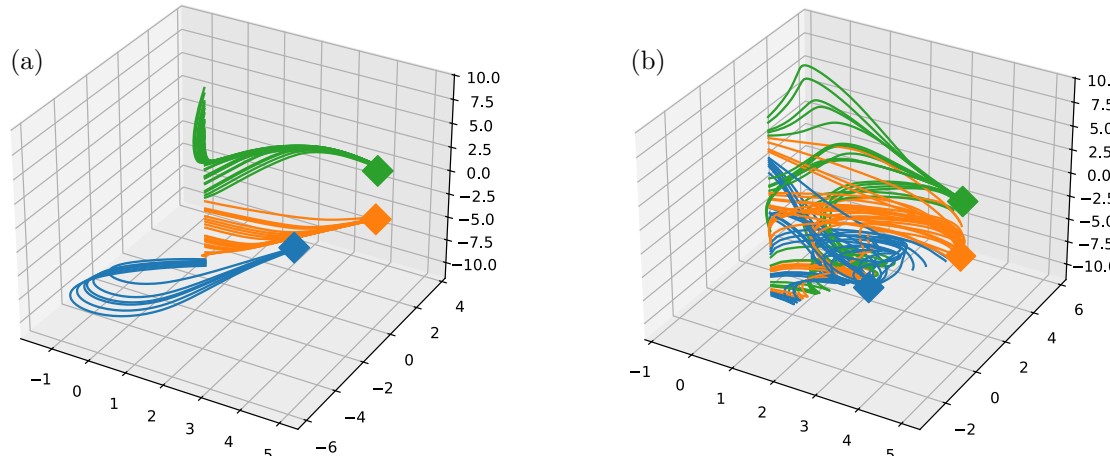

Figure 10: Classifier flow lines of the radial (a) and angular (b) classification problems. Colours indicate which label the trajectory belongs to and what label the target points (squares) correspond to. The latent representations constituting the classifier input correspond to the line where the integral curves start.

## B   Experimental Details

### B.1   Reconstruction Experiments

All neural networks are sequential with four fully connected layers of width 200 and tanh as activation function. The first three layers have bias, the last one does not. The biases are initialised as 0 and linear layers are initialised according to a normal distribution. For the cut-off in the custom backpropagation, we use $b = 10^{-7}$, and $c = 10^{-6}$. We recall that this introduces an error in the compressed directions of at most $b$.

Using the Euler ODE solver implemented in the `torchdiffeq` Python package with 500 time steps, along with the adjoint method for the backpropagation of the same package, (Chen, 2018), we solve the flow equation in two stages, each integrating for half the total time. Thus, after the first stage, the flow maps the input data into the latent space. There, we project to the latent space to eliminate the remaining numerical error in the latent components, which we find to be of order $10^{-8}$, then we continue the flow. The models are trained using RMSprop, with momentum 0.3.

The experiments were run on a cluster, using the CUDA backend of PyTorch with a NVIDIA Tesla T4 GPU, 16GB VRAM, and Intel(R) Xeon(R) Gold 6226R CPUs (2.90GHz). The replication experiments for the spiral with 3 turns over different seeds were run with an NVIDIA RTX 4090 GPU and an and Intel(R) Core i9-14900K CPU.

**Spiral**   For the spiral experiment, consider the following parametrisation where $v \in \mathbb{R}$, $e_2 = (0, 1, 0, 0)$.

$$f : [0, 2\pi] \to \mathbb{R}^3, \quad s \mapsto (1 + 0.5s)(\cos(vs), \sin(vs), 0) + 6e_2.$$

We vary the number of turns, $N$, from $N = 0.5$ to $N = 5$ by changing the speed $v$. The input training data is equidistantly sampled according to the intrinsic distance $d_S$ of the spiral, which can be computed analytically. The number of sample points is 20 times the length of the spiral, rounded down, and capped at 5000 due to memory constraints during training. For the validation data, 0.3 times the number of training samples are drawn randomly in the angle domain $(0, 2\pi)$.

The models are trained for about 27000 epochs unless the loss does not improve significantly. This happened only for $N = 0.5$ and $N = 1$, which stopped at about 8000 and 18000 epochs, respectively.

**Sphere**   Consider a two dimensional round sphere of radius 1 in $\mathbb{R}^4$ around the point $x_0 = (0, 0, 0, 3)$ given by $\{(z_1, z_2, 0, z_3) \mid (z_1, z_2, z_3) \in S_1^2(x_0) \subset \mathbb{R}^3\}$. The $z \in S_1^2$ in turn are parametrised by spherical coordinates. The training data is sampled from the sphere, equidistantly on a grid in the angle domain $(0, 2\pi) \times (-\pi, \pi)$

with 6000 points in total. The 1800 validation data points are sampled randomly from the angle domain. The model was trained for about 10000 epochs.

## B.2 Classification Experiments

Both the radial and angular classification experiments use the same neural network architecture. The layers consist of a linear input layer with input size 4 (spatial + time) and output size 512, followed by two hidden layers with input and output size 512. A skip connection is used between the hidden layers. The final output layer is a linear layer with input size 512 and output size 3. All layers are initialised using Xavier uniform initialisation (Glorot & Bengio, 2010), and the hidden-layer biases are zero-initialised. ReLU is used as the nonlinearity between layers. The models are trained using the Adam optimiser. A reduce-on-plateau learning rate scheduler is used with a patience of 20 epochs and a factor of 0.8. Both experiments run for 300 epochs, with a batch size of 32.

The target points are placed equidistantly to the latent line at the points $(-3, 4, 0)$ for $l_1$, $(0, 5, 0)$ for $l_2$, and $(3, 4, 0)$ for $l_3$. For the attractor terms in (11) we use $C = 50$ and $k = 64/d_{\min}$, where $d_{\min}$ is the minimum distance between any two target points ($\sqrt{2}$ in this setup).

Experiments were conducted on the CUDA backend of PyTorch with an NVIDIA RTX 4090 GPU, with CUDNN set to deterministic mode and a fixed random seed of 42, and Intel(R) Core i9-14900K CPU.

