# OpenReview forum: "PolyNODE: Variable-dimension Neural ODEs on M-polyfolds"
_TMLR — Under review for TMLR_

### Review · Reviewer_ebVo · 2026-06-27

**Summary Of Contributions:**

This paper proposes PolyNODE, a neural ODE model generated by learnable semi-flows defined on M-polyfolds. An M-polyfold is an extension of a manifold that allows intrinsic variable dimension while still supporting differentiation. Therefore, PolyNODE can be understood as an extension of conventional fixed-dimensional neural ODEs to a setting in which the hidden dimension can vary, as in ordinary neural networks. In this sense, PolyNODE can be viewed as a continuous-depth limit model that naturally represents structures with bottlenecks, such as autoencoders.

Neural ODEs embedded in a high-dimensional ambient space can also imitate a similar architecture. However, the authors point out that, in a strict sense, such models cannot enter a genuine bottleneck in finite time. If one attempts to enforce this, finite-time collapse requires the dynamics to become singular, and the differentiation and backward adjoint training used in standard smooth NODEs are no longer justified under such singularities. This motivates the need for PolyNODEs, which can enter a bottleneck structure in finite time while retaining a notion of differentiability.

To this end, the authors first define sc-Banach spaces, along with sc-continuity and sc-derivatives on them. Intuitively, this introduces a layered structure of increasing regularity on a function space, allowing the singular collapse that occurs when dimension decreases to be treated as smooth scale-wise vanishing. This structure makes it possible to discuss differentiation, tangent space, and the chain rule near the bottleneck. A M-polyfold is then defined as a space constructed by gluing retract images in an ambient sc-Banach space. Here, a retract is a projection-like operator that preserves transverse directions outside the bottleneck and sends them to zero inside the bottleneck. The authors explain how the simplest M-polyfold, $\Omega_1^1$, which has $(1+1+\text{time})$-dimensional structure, $(1+\text{time})$-dimensional bottleneck, and then again a $(1+1+\text{time})$-dimensional structure, can be implemented in the sc-Banach space $\mathbb{R} \oplus \mathbb{R} \oplus L^2(\mathbb{R})$. They then extend this construction to $\Omega_n^m$, which has an $(n+m+\text{time})$-dimensional structure, an $(n+\text{time})$-dimensional bottleneck, and then again an $(n+m+\text{time})$-dimensional structure, thereby successfully introducing the relevant polyfold geometry.

The authors then discuss how to define ODEs on M-polyfolds, by formulating vector fields on the underlying sc-Banach spaces and requiring them to be compatible with the retract structure. In the case of a sc-Banach space without a bottleneck, if the solutions generated by the vector field on each scale are mutually consistent, an appropriate sc-flow can be obtained. However, in the presence of a bottleneck, the flow loses injectivity at the bottleneck and therefore cannot be a flow in the usual locally invertible sense. The authors thus introduce semi-flows, meaning flows that are well-defined only forward in time. Furthermore, they construct finite-time compression vector fields so that, under such semi-flows, the $m$-dimensional transverse coordinates vanish sufficiently rapidly, ensuring that the trajectory remains in $\Omega_n^m$. In this way, they construct a flow that is sc-continuous and realizes a genuine bottleneck.

Using this polyfold ODE framework, the authors finally define PolyNODEs. In essence, a PolyNODE is a semi-flow generated by learnable finite-time compressing vector fields on an appropriate $\Omega_n^m$ within an appropriate sc-Banach space. These learnable compressing vector fields are explicitly defined in Eq. (9), ensuring sufficiently rapid finite-time transverse collapse. However, although the forward dynamics of such a learnable semi-flow is well-defined, training it with a backward adjoint faces a derivative blow-up at the bottleneck due to the non-Lipschitz nature of Eq. (9). To address this, the authors introduce a cut-off function into Eq. (9), which turns off the compressing vector field when the $m$-dimensional transverse coordinates become sufficiently close to zero. In the backward pass, they then use a custom derivative corresponding to this cut-off-regularized vector field. Thus, in practice, the implemented PolyNODE corresponds to a cut-off surrogate of the ideal compressing Eq. (9).

The authors then experimentally demonstrate, on relatively simple synthetic objects such as spirals and spheres, that the proposed PolyNODE-based continuous-depth autoencoder can compress and reconstruct the underlying geometric structure.

**Audience:**

Yes

**Audience Explanation:**

I expect that at least a subset of the TMLR audience would find this paper interesting, especially researchers working on flow-based and continuous-depth models, geometric deep learning, and the mathematical foundations of deep learning architectures. The use of M-polyfolds is novel in the ML context and may open a new conceptual direction for studying continuous-depth analogues of variable-width neural networks.

That said, the current paper is more of a mathematical and conceptual contribution than a fully developed practical ML method. Readers primarily interested in scalable architectures or benchmark performance may find the experimental section somewhat limited.

**Broader Impact Concerns:**

I do not see any specific ethical implications of the work that would require further discussion.

**Claims And Evidence:**

Yes

**Claims Explanation:**

The main conceptual claim, namely that standard NODEs are intrinsically fixed-dimensional whereas semi-flows defined on M-polyfolds provide a possible mathematical framework for variable-dimensional continuous-depth dynamics, is well motivated and clearly explained. The paper provides explicit constructions of bottleneck spaces and vector fields that traverse them, which supports the central geometric idea. Overall, I think the core claims are supported by a clear and interesting construction.

That said, the empirical evidence is still preliminary. The experiments are convincing as proof-of-concept demonstrations, but they are limited to synthetic geometric examples. There is no quantitative comparison with standard autoencoders, augmented NODEs, ambient-space NODEs, or other relevant baselines. Therefore, the experiments support the feasibility of training the proposed construction, but they do not yet establish its practical advantage or necessity.

Moreover, a potential issue concerns the numerical treatment of the derivative blow-up that arises when training the compressing vector field in Eq. (9). The implementation handles this through cut-off regularization. While this is a reasonable numerical strategy, the paper would be strengthened by a more explicit discussion of the status of this issue. In particular, it would be helpful to explain whether the blow-up is specific to the particular choice of Eq. (9), or whether it is a more general issue shared by any class of vector fields that achieves sufficiently rapid finite-time transverse compression. My impression is that any vector field satisfying the latter condition may face a similar non-Lipschitz or derivative blow-up issue, so a clarification on this point would strengthen the paper.

**Requested Changes:**

- The paper would benefit from a more detailed discussion of the cut-off-based surrogate used during training. In particular, it would be helpful to clarify whether this numerical bypass is generally required for a broader class of compressing vector fields, or whether it mainly arises from the specific parametrization adopted in Eq. (9). If some form of relaxation is indeed necessary for training arbitrary finite-time compressing vector fields, the authors may wish to explain how such a mechanism could be implemented in a more general setting. It would also be useful to discuss the potential discrepancy between the surrogate flow used during training and the ideal exact-collapse semi-flow at test time.

- If possible, the paper would benefit from comparisons with the ambient-space NODE models mentioned in the text, such as augmented NODEs and AutoencODEs. I understand that the main goal of the paper is to introduce PolyNODE as a new deep learning architecture and to provide a proof of concept. Nevertheless, since the experimental evaluation is currently somewhat limited relative to standard ML papers, it is not yet easy to assess the practical advantages of the proposed formulation. Even lightweight comparisons on the synthetic reconstruction tasks could help illustrate when the polyfold-based approach provides advantages over existing NODE-based alternatives.

---

> ### Author Response · Authors · 2026-07-21
> **Review response**
>
> Thank you for your very careful review of our paper and for identifying important points that deserve further elaboration. Below, we address the requested changes and specify the revisions to the manuscript we have made in response to them.
>
> **Requested changes:**
> * Thank you for this suggestion. We agree that the numerical implementations - specifically, the cut-off functions used to regularise the vector fields - should be discussed in more detail and have modified Section 3.3 to elaborate on the points. In particular, we emphasise the fact that the regularisation is not a feature of the specific class of compressing vector fields we use in the numerical experiments. Any non-Lipschitz vector field will have a similar singularity at the stratification that needs to be regularised in order to employ standard gradient-based methods. (See also the response to reviewer JVuq).
>
> * We agree that it is not easy to assess the practical advantages of the PolyNODEs over ambient space NODE models based on our numerical experiments. In fact, for the simple geometric tasks considered in the paper, we don't expect there to be an advantage in terms of the reconstruction loss. The advantage is rather principled, in that the PolyNODEs provide a theoretical model that retains a notion of differentiability in the presence of the stratifications. In the regularisation of the vector field, we maintain explicit control of the transverse error, which would not be possible in an ambient-space approach with a loss term enforcing an (approximate) dimensional reduction. That being said, we are cautious to include numerical comparisons of this error since the ambient space models are not trained to minimise it (or indeed even define it). For example, the AutoencODE model explicitly disregards the singularity at the stratification and introduces auxiliary ambient space dimensions to prevent the transverse error from influencing the dynamics after the encoding. We have added a paragraph to the Conclusion discussing numerical and practical aspects of the comparison to ambient space models.

---

### Review · Reviewer_SihQ · 2026-07-02

**Summary Of Contributions:**

This work uses M-polyfold theory, which informally speaking, allows for smooth derivatives between spaces of different dimensionalities that have been "glued" together, to extend neural ordinary differential equations (NODE) to learn flows that vary in dimension over time. The key to doing this is a retraction map over a scale Banach space (defined using "scales" which are a series of increasingly restrictive sets) that defines local regions (retracts) on which derivatives are well-defined regardless of dimensionality in the original space. This work explicitly defines a scale in terms of weighted Sobolev norms and a retraction map over the space $\Omega_n^m$, which is a $1+n+m$-dmensional space that goes through a "bottleneck" of $n+1$ dimensions, proving that they satisfy the criteria required to be a M-polyfold. Finally, NODEs are trained on this space as continuous analogues of an autoencoder, learning flows that pass through the bottleneck to reconstruct data in the shape of a spiral or sphere.

The paper is well written and has an impressive summary of how differential geometry is constructed over M-polyfolds. The experimental setup is as clean as can be made for such a complex concept, and the results are promising although there are some issues (see below).

**Additional Comments:**

What happens to points not in the data manifold (i.e. outside the spirals/spheres?) when passed through the trained semi-flow? I ask because the autoencoder objective enforces bijection, but the semi-flow is not injective generally speaking.

**Audience:**

Yes

**Audience Explanation:**

This paper will naturally be of interest for NODE researchers and applied mathematicians working with ODEs. The work presents a nice and concise summary of M-polyfold theory with an explicit construction and so will be a helpful introduction to the area for those outside of the field. It is also a necessary prerequisite to the follow-up work discussed in the conclusion, namely dynamics which are not possible in the ambient space of a bottlenecked space $\Omega_n^m$.

My main suggestion is, in order to make this work accessible to an even wider audience of machine learning researchers, the exposition in section 2 (particularly section 2.1) could be elaborated further with intuitions and informal explanations. The reason for this is that for someone not well versed in ODEs, M-polyfold theory is at least twice-removed from one's knowledge base, making the notation and derivations naturally hard to follow. See requested changes for details.

**Claims And Evidence:**

Yes

**Claims Explanation:**

The main claims are that it is possible to apply M-polyfold theory to train NODEs over spaces with varying dimensionality. This is clearly shown both in the explicit derivations and the experiments that show an autoencoder can be trained, albeit on toy data. There are two significant issues with the experiments, but given that this paper is a combination of theory and proof-of-concept, the derivations are sufficient to support the claims.

The empirical issues are:

1. The training procedures appear fragile and inconsistent - specifically, the need for 2 regularization terms additional to reconstruction loss (section 4.1), and $L_3$ in particular (encouraging the flow to be approximately isometric) is a pretty strong and potentially task-specific condition. Perhaps one way to show that these conditions are more general and not problem-specific would be to decode into a different shape, instead of reconstructing the same spiral. It is also unclear how to choose other hyperparameters such as the start/end times $\tau_\mathcal{X}$ and $\tau_\mathcal{Y}$
2. There is some uncertainty about whether the autoencoders are actually compressing the data manifold. See requested changes for details.

**Requested Changes:**

Two significant issues with the experiments:
- The statement "these plots also show that our autoencoding NODE achieves very good encoding even when the reconstruction suffers slightly" is overclaiming because the compression step guarantees that the encoding will look correct (like a line) even if there are some "kinks" due to poor encoding. It would be more convincing to show the monotonicity error (same as figure 4a) after training for each spiral. Similarly, it would be best to show the reconstruction error for each spiral, maybe in a table.
- sphere reconstruction (section 4.2) places one of the sphere's dimensions in time (first element). This has some problems: a) although figure 7 is visually attractive, it is more confusing to actually parse what is going on, b) it feels like since the flow is still 3-dimensional through the bottleneck there is no compression required to encode the 2-sphere, c) figure 7 says it plots time slices but due to the embedding of the sphere in the time dimension, these are not really time slices as the points come from multiple time steps, so it is not clear how the points are actually moving through time. It would be best to have another version of this experiment where the sphere is embedded in the same way as the spiral (i.e. without affecting the time dimension).

Minor points that could use more elaboration in the exposition:
- noting the significance of $E_1$ vs $E_0$ in equation 3 (maybe the first sentence on page 5 should be moved earlier?)
- explaining the intuition behind $\beta$ as a shift parameter and how the retraction defined in equation 4 handles the change in dimension towards $\tau_1$ and $\tau_2$
- it was not clear to me how the inequalities at the bottom of page 4 (following the definition of the retraction $r$ equation 4) were derived: specifically, does the term $\exp(\delta_k \beta(\tau))$ come about from shifting $s$ in $\exp(\delta_k s \alpha(s)$, and how does the negative sign come about in $|\langle f, \gamma_\tau \rangle | \leq  C  ||f||_{k, I \tau} \exp ( -\delta_k \beta(\tau))$?
- maybe note that the third term of the differential $Dr$ tens to 0 near $\tau_1$ and $\tau_2$ and that it i s defined to be 0 at $\tau_1$ and $\tau_2$ (relevant to first equation at top of page 5)
- is it necessary for the third component of $Dr$ to converge to 0 in each of the $k$ scales, or can it just be finite (according to the definition in equation 2)?
- maybe some sub-headings under "$\Omega_1^1$ as an M-polyfold" would help make clear what the goal of each part is, i.e. everything up to "Taking the $\tau$-derivative [...]" before equation 5 is  related to showing $r$ is a retraction, whereas everything after relates to tangent spaces
- maybe explicitly state somewhere when using subscript i to indicate the ith element, such as in $x_1$ "A latent representation of a point $p \in \mathcal{X}$ is given by $\phi(t, p)$ for any $t$ such that $\phi(t, p)_1 \in [\tau_1 , \tau_2]$" (section 3.1)
- explaining the intuition behind the choice of cutoff function in section 3.3: in particular, why a 3rd order polynomial, and what are the endpoints $b$ and $c$ set to?
- it is not always clear which regions are parameterized by neural networks: namely "We then obtain a family of compressing vector fields by parametrising the functions $k_j$ using neural networks." (top of page 8) combined with equation 9 seems to imply that the semi-flow in $[\tau_0, tau_1]$ is determined by the neural network, but table 1 suggests that the semi-flow is only being compressed in the extra $m$ dimensions and no parameterization is occuring. Similarly I am not sure if it is explicitly stated that the latent region ($\tau_1$ to $\tau_2$) is also unparameterized.
- define $d_S$ earlier where $c_d$ is defined (section 4.1)

Other minor points regarding the presentation:
- Is the x-label in figure 4b ("speed") wrong? The caption says it is spiral diameter.
- although the discussion touches on some potential directions where PolyNODE has advantages over prior methods that use an ordinary NODE (e.g. Cipriani et al. 2025), it would also be helpful to discuss some of the disadvantages and general differences between these works and PolyNODE.

---

> ### Author Response · Authors · 2026-07-21
> **Review response (1/2)**
>
> Thank you for your very thorough review of our paper. We are grateful for both the insightful constructive critique and for your many helpful suggestions for improving the quality and presentation. Below, we address the questions raised and requests made, point by point, and describe the revisions we have made in response to them.
>
> **Major issues:**
> * We report a bound for the monotonicity error after training for all spirals at the end of the paragraph were we introduce the monotonicity error. "All experiments show very good monotonicity on the sampled data points, with fewer than $0.02\%$ misaligned points."  We have added the plots for the monotonicity error for three more spirals and rewritten the paragraph to make our claim more precise and better supported by our numerical results. The relative reconstruction error for all spirals was shown figure 4b, but the figure was mislabelled. The x-axis should have read "number of turns". We have promoted the reconstruction error vs. number of turns to a separate figure and corrected the mistake in the caption.
>
> * It is important to distinguish between the flow time $t$, and the parameter $\tau$ in the retract construction. This also resolves the confusion about the time slices. Time slices, like the latent representation defined in the last paragraph of section 3.1, are taken with respect to the flow time. Due to the setup for the spirals we have $\phi(t, \mathcal{X}) \subset \{ (\tau, x,y) \mid \tau = \tau_\mathcal{X} + Ct \}$ in that case. For the embedding of the sphere into $\\mathbb{R}^4$ this is not the case any more, but the (deformed) spheres in figure 8 are still flow time slices. Regarding the compression in the sphere example: The goal is to extract a latent representation of a 2-sphere. Since the presented model is based on $\Omega^m_n$, the latent space is $[\tau_1, \tau_2] \times\mathbb{R}^{n}$, and $\mathbb{R}^3$ is the smallest Euclidean space that admits an embedding of a 2-sphere. Thus, the flow is necessarily three-dimensional in the latent region. For $\tau<\tau_1$ and $\tau>\tau_2$, however, it is four-dimensional. The last coordinate is compressed, and the embedding of the sphere uses this coordinate. We have updated section 4.2 to clarify the above points.
>
> **Minor points regarding exposition:**
> Thank you for the many concrete suggestions for improving the exposition of the paper. We have implemented them all, and specify the details of the changes below. In connection to the first two points, we have also added more intuitive explanations in section 2.1 as requested in the main review text.
>
> * We made this point more explicit after eq. (3).
>
> * We added a couple of sentences illustrating the intuition about $\gamma_t$ and $\beta$'s roles for the dimensional collaps after the definition of $\gamma_t$.
>
> * We have added a short derivation of the two estimates in question.
>
> * We have rewritten the paragraph following the definition of $Dr$ to point out that the third term of $Dr$ tends to 0, and to clarify the following point.
>
> * According to the definition of scale continuity in the second paragraph on page 4 the restricted maps $Dr|_{(E^1\oplus E)_k} : E_k+1 \oplus E_k  \to E_k$ must be continuous with respect to the respective topologies; that is at $\tau_1$ and $\tau_2$ the map $Dr$ must go 0 in every scale.
>
> * We have restructured section 2.1 and added new subheadings as suggested.
>
> * We have added an explanation for subscripts after equation (6).
>
> * We have revised Section 3.3 to include a motivation for the particular cut-off function used. We have also extended the discussion of the regularisation (see also the review by JVuq, and our response to it). The choice of the coefficients $b$ and $c$ is specified in Appendix B.
>
> * This is true, the general autoencoding PolyNODE model we present in Sections 2 and 3 allows for parametrised compression rates $k_j$ and latent dynamics. This complexity was not necessary for the examples we present in Section 4. The paragraph of Section 4 has been changed to clarify this.
>
> * We have moved the definition of $d_S$ as suggested.
>
> **Other minor points:**
> * Thank you for noticing this mistake. We have corrected the horizontal axis label and the caption. Both should, in fact, be number of turns for the spiral.
> * Thank you for requesting a discussion of the drawbacks of the PolyNODE models compared to other architectures. We completely agree that this is an important aspect missing in the presentation. We have added a paragraph in the discussion to clarify the disadvantages of our approach compared to other methods.

---

> ### Author Response · Authors · 2026-07-21
> **Review response (2/2)**
>
> **Additional comments:**
> Thank you for raising the questions regarding how the semi-flow acts on points off the data manifold. It is correct that the semi-flow is not injective. However, how this affects points off manifold depends on the precise relationship between the dimensionality and topology of the data manifold and the latent space. If the latent space is matched to the data manifold, the compression will necessarily distort points off-manifold and fail to reconstruct them. This is similar to a traditional autoencoder, which is also not injective on the entire input space.

---

### Review · Reviewer_JVuq · 2026-07-12

**Summary Of Contributions:**

The paper asks a natural question: what does a neural ODE look like if the state dimension is allowed to change over time? Since manifolds can't support this, the authors use M-polyfold theory, which gives stratified spaces a workable notion of smoothness. They build an explicit M-polyfold structure on a bottleneck space with weighted Sobolev scales. On top of this, they define PolyNODEs (maps given by semi-flows of vector fields on the retract), construct an autoencoder using compressing vector fields that reach the bottleneck in finite time, and adapt backpropagation with a cutoff to handle the derivative singularity. Experiments show the model can autoencode spirals and a sphere, and that the latent representation supports simple downstream classification.

**Audience:**

Yes

**Audience Explanation:**

Yes. The question of how to define continuous-depth models whose state dimension changes over time is natural and, to my knowledge, has not been addressed intrinsically before. The contribution is mathematically substantive, and will interest researchers working on neural ODEs, flow matching, and geometric deep learning, as well as those interested in the mathematics of stratified spaces in machine learning.

**Broader Impact Concerns:**

None. This is foundational, theoretical work on the geometry of continuous-depth models, validated on synthetic toy data (spirals, spheres).

**Claims And Evidence:**

Yes

**Claims Explanation:**

The paper makes two modest claims and mostly backs them up: (a) Neural ODEs can be generalized to "M-polyfolds," which lets the model change its own dimension as the dynamics run. (b) These models can be trained for reconstruction using a modified version of backpropagation.

The support is solid. Rather than just asserting that the map defining the polyfold is smooth (in the "sc" sense), the authors verify it with explicit estimates. The claim that the vector fields compress in finite time follows from standard finite-time stability results. And the experiments show the models actually train.

The one real weak spot is that the theory and the implementation don't quite match. The formal definition requires the vector fields to be $sc^k$-smooth, but the fields doing the compression are only Hölder-continuous at the bottleneck — a weaker condition. And the code departs further still: it uses a smoothed/cutoff version of the field and forces the state onto the latent stratum partway through the integration.

**Requested Changes:**

- Add a subsection stating clearly the regularity class of the trained vector fields

- Collect reconstruction and monotonicity errors per number of turns N, and the sphere reconstruction error, with results over multiple seeds.

---

> ### Author Response · Authors · 2026-07-21
> **Review response**
>
> Thank you for your careful review of our paper and for suggesting ways to clarify our theoretical construction and improve the presentation of our experimental results. Below, we address the two requested changes and comment on the mismatch between theory and implementation discussed in the evaluation.
>
> **Theory/implementation mismatch:** Thank you for raising these points. The first point concerns the theoretical construction of our compressing vector fields. It is correct that the vector field $Y$ on $\Omega^m_n \subset \mathbb{R}^{m+n+1}$ is Hölder-continuous. However, this is no obstruction to the lifted vector field or the semi-flow being $\mathrm{sc}^0$. We have revised section 2.2 and 3.2 to emphasise these points.
>
> The second point is that the smoothing/cut-off used in experiments affects the regularity of the vector fields. This is indeed correct. We have clarified section 3.3 to explain this regularisation more clearly. We have also extended the discussion in 3.3 to include the impact of the regularisation on the dynamics in the directions transverse to the latent space, and how this is controlled through the coefficients $b$ and $c$ appearing in the cut-off function. This was previously only discussed in Appendix B on experimental details. We have also changed the name of section 3.3 to emphasise that it concerns regularity in the implementation as requested. See also the response to reviewer ebVo.
>
> In fact, any non-Lipschitz vector field will require similar regularisation at its singular points in order to apply standard numerical gradient-based learning methods.
>
> **Requested changes:**
>
> * We have extended section 3.3 to include a more detailed discussion of the regularity in implementations. We felt this would be more easily accessible to the reader than deferring the discussion to a separate subsection.
>
> * We have included additional plots of the monotonicity error for more numbers of spiral turns (see also the response to reviewer SihQ). Due to our limited computational resources, running multiple seeds for all our experiments during the discussion period is unfortunately not feasible. We have, however, run one of the spiral ($N=3$ turns) reconstruction experiments for five different seeds, and included the results in the updated version of the manuscript. We observe no significant influence of the initialisation on the results (neither monotonicity nor reconstruction).

---

### Author Response · Authors · 2026-07-21
**Revision uploaded**

We wish to express our sincere gratitude to the reviewers for their work, insightful feedback, and many constructive suggestions for improving our manuscript. We have uploaded a revised version of the manuscript based on the reviews and added responses to the individual reviewers below. Thank you!

/The authors